# DIAGRAMMATIC SUMMARIES
# FOR NEURAL ARCHITECTURES

**Guy Clarke Marshall**[1]**, Caroline Jay**[1] **& André Freitas**[1,2]
Department of Computer Science[1]
University of Manchester
Oxford Road, Manchester, M13 9PL, UK
Idiap Research Institute[2]
Rue Marconi 19, Martigny, 1920, Switzerland
`guy.marshall@postgrad.manchester.ac.uk,`
`{caroline.jay, andre.freitas}@manchester.ac.uk`

## ABSTRACT

This paper advocates for diagrammatic summary publications for machine learning system architecture papers. We review existing diagram-centric scholarly practices, and summarise relevant studies on neural network system architecture diagrams. We subsequently propose three opportunities: Diagram guidelines, diagrammatic system summary publications, and the community creation of a formal diagram standards, which could be integrated with existing LaTeX + PDF publication processes.

## 1 INTRODUCTION

This position paper focuses on improving communication about Machine Learning (ML) systems within the existing LaTeX + PDF scholarly publication process. We advocate for use of static diagrams as a more efficient medium. Three proposals result:

1. *Diagram guidelines* proposes support to help authors make existing diagrams (which authors are already creating) better.

2. *Diagrammatic system summary* proposes a new publication format, within the existing PDF + LaTeX and peer review publication model.

3. *Diagram standards* proposes a community effort to create a standard to maximise the efficiency of communicating visually describe the essential content of ML system research.

These proposals are designed pragmatically, potentially for sequential implementation. Through guidelines, an empirical basis for good practice is established. In system summaries, the quality of guideline-compliant diagrams is refined through peer-review, with authorship benefits and further exploration of good practices. Diagram standards based on lessons from these formalise diagrams with precision and completeness of content, with an accompanying cognitive benefit to communication provided by a visual language.

### 1.1 MOTIVATION

At present, comparison between different methods or different architectures is cognitively challenging, and simply keeping up with the state of the art is a challenge. More efficient communication practices would reduce the effort for research communication, and reduce barriers to accessing knowledge.

As highlighted in the call for this workshop, "disseminating knowledge to the next generation of scientific researchers" is a challenge. In industry, as well as in academia, software architecture diagrams are often used to debate ideas and to communicate to current and future employees about existing processes, systems and decisions made. We propose further emphasis, bringing diagrams

to the centre of communication about systems, and improving the quality of static diagrams describing AI systems. In this way, many architectural contributions could be summarised through a single system diagram. Further, consistent diagrams have the potential to increase reproducibility, clarity, comparability, accessibility and reduce ambiguity. These attributes are essential to efficient communication.

Readers of new machine learning architecture papers will be familiar with some of the different diagram formats used to describe systems. A collection of these diagrams has been made available at aidiagrams.com. Across different architectural paradigms such as CNNs and BiLSTMs, there are commonalities in their visual representation, there is little overall consistency. No two diagrams in major conferences have the same visual representation. The proposals here aim to shepherd this heterogeneous diagrammatic landscape towards being more effective, both for readers and authors. The approach is pragmatically designed to be more incremental than revolutionary. However, we believe this approach has significant benefits in terms of feasibility and implementation, and could be complementary to other methodological improvements such as making code more readily available (Pineau et al., 2020).

The discussion here only applies to contributions where an application or system architecture is a main contribution of the paper. This holds for the majority of papers at the major Natural Language Processing and Computer Vision conferences, for example ACL, CVPR, and other application-specific venues, as well as for architectural papers at NeurIPS and other neural network-centric ML venues.

## 2 BACKGROUND

### 2.1 BENEFITS OF DIAGRAMS

Diagrams are useful for describing systems for their spatial-relation advantage (Karaca, 2012) and explanatory value (Burnston, 2016). Diagrams:

- Are external representational support to cognitive processes (Clark & Chalmers, 1998).
- Make topics simpler, leading to reduced search space and fewer cases to be computed over, by including minimal salient information (Sloman, 1984).
- Are manipulated in order to profile known information in an optimal fashion (Tylén et al., 2014).
- Make abstract properties and relations accessible (Hutchins, 1995).
- Facilitate perceptual "free rides" in inference (Shimojima, 2015).
- Can be in a public space, therefore enabling collective and temporally distributed forms of thinking (Peirce, 1931–1966).

These attributes make diagrams an appropriate medium for scholarly reasoning and communication. They have the potential to be a much more efficient way than text to understand and research ML systems. In practice, they are also prevalent, suggesting authors also feel diagrams are an appropriate medium for communicating about these systems.

### 2.2 EMPIRICAL DATA ON ML SCHOLARLY DIAGRAM USAGE

A recent analysis of all papers at ACL, a top Natural Language Processing conference, found that system diagrams are often used by authors (Marshall et al., 2021a). 160/195 (82%) of all ACL 2017 papers included diagrams to represent concepts (not results or algorithms). 124/195 (64%) of all ACL 2017 papers included at least one system diagram (119 of these were neural architectures). Given the 8 page limit at ACL, the frequent inclusion of system diagrams, usually requiring half a page or more, shows the importance authors place on system diagrams for communicating their research.

A recent study by Marshall et al. (2020) reports on diagram usage by ML researchers in reading and writing papers. In this study, all participants reported using diagrams to get a summary of the paper. They also reported some participants giving primacy to the diagram over all other text, with 3/12

participants "reading the diagram before the text of a scholarly publication". The findings suggests diagrams have a central role in scholarly communication about ML systems. The identification of this crucial role of diagrams in communicating about AI systems leads us to propose two different opportunities which have been identified to improve scientific dissemination about Machine Learning, without amending the existing publication processes.

Diagrams describe the core abstractions and representational and algorithmic dimensions of the system, and may contain information about the task and the application. If one adds some key metrics, they can also show the results for that system, within peer-reviewable evaluation standards (which also provides evidence of methodological rigour). In their present form diagrams do not facilitate comparison, and therefore cannot communicate the difference with existing related work. In their present form of heterogeneity, many of the diagrams used have high risk of miscommunication, and of communicational inefficiencies.

## 2.3 VISUAL LANGUAGES AND GRAPHICAL ABSTRACTS IN OTHER DOMAINS

Formal visual languages are used in many scholarly communication practices. Graphical abstracts also integrate, usually without being prescriptive in content or visual encoding, diagrams into the existing publication model.

### 2.3.1 VISUAL LANGUAGES FOR TECHNICAL SYSTEMS

There are various diagrammatic languages used to communicate about software. Unified modelling language exists (Rumbaugh et al., 2004), as does SysML and many others. In practice these are rarely (if ever) seen at top Natural Language Processing, Computer Vision or Machine Learning venues. Given their fame and wide pedagogical usage, we assume the authors know of these diagrammatic languages but choose not to use them, perhaps because they do not provide suitable visual elements for ML functions, are not succinct for expressing complex relationships, and do not provide visual aids for relevant structural change of data.

In high energy particle experiments at CERN, a study of their communication diagrams suggested that "diagrams are more appropriate than texts to represent the procedural information necessary" (Karaca, 2017). This may also be true in ML.

"Synthetic biology open language visual" (SBOL Visual 2.0) is an established visual language for scholarly communication (Beal et al., 2019). SBOL Visual 2.0 is comprised of a set of glyphs, arranged in a standardised grammar, with an accompanying specification, ontology and examples, made publicly available. It is centrally curated, with users able to submit requests for changes such as new glyphs. van der Linden et al. (2019) note that "The main notations used by practitioners (UML, BPMN, SysML, ArchiMate) are regulated by standardization bodies." It may also be useful for visual languages for ML to be curated centrally and learn from established standardization bodies.

Penrose (Ye et al., 2020) is a system for creating mathematical diagrams, converting mathematical notation into visualisation representations. Examples of their domain include sets, functions, geometry, linear algebra, meshes and ray tracing.

### 2.3.2 GRAPHICAL ABSTRACTS

Graphical abstracts (GAs) are a component of some research articles which aim to visually "communicate the essential features of an article". In their guide for research presentation, Patience et al. (2016) favour images over text in GAs: "communicate essential features of research with multiple images and some text" and state "GAs must be self-explanatory".

The empirical data and our experience of communicating through the ML paper medium suggests that utilising system diagrams as GAs may be appropriate for ML. There are, however, concerns about graphical abstracts. Pferschy-Wenzig et al. (2016) found that graphical abstracts are associated with *reduced* visibility to scholarly publications. The study examined a single publication venue, "Molecules", over a limited timeframe. As noted by the authors, it does not provide conclusive or generalisable evidence against graphical abstracts.

In their analysis of 54 GAs, Hullman & Bach (2018) define a taxonomy to describe, classify and analyse the visual structure of GAs, noting "design of GAs is more diverse in its use of spatial

layout than the textbook diagrams, which were presumably created by professional artists". Larson et al. (2017) edit 50 scientific GAs according to design principles, and show that "redesign improves understanding of the paper". These studies suggest there are benefits in utilisation of visual design principles, rather than being supportive of the GA artefact itself.

### 2.3.3 NANOPUBLICATIONS

Nanopublications are a Linked Data format used for scholarly data publishing, most commonly in Life Sciences (Kuhn et al., 2018). Being an RDF format for each item, and capturing provenance, the linked data is optimised for machine reading at scale, rather than human interpretation. In our diagrammatic system summary position, we advocate a similarly ontologically unambiguous stance, but optimised for human interpretation. This could be integrated with an existing data model such as PROV-O (Lebo et al., 2013), so that AI system architectures are expressed in an linked data format, allowing credit to be attributed at an atomic system-component level. However, at this stage we avoid integration with a provenance-based nanopublication model, in order to focus attention on the efficacy of diagrams for communicating about ML.

## 3 POSITION PROPOSAL

### 3.1 POSITION 1: DIAGRAM GUIDELINES

The first proposal is lightweight, in that it can be implemented using the existing publication skills, processes and artefacts. Our argument for diagram guidelines is as follows:

- Most authors use system diagrams (Marshall et al., 2021a).
- There has been little attempt to define or adopt guidelines or standards for neural network system diagrams.
- At present, system diagrams are hetereogeneous, both in content and visual encoding (Marshall et al., 2021a).
- This is having negative consequences for readers (Marshall et al., 2020), and is not an optimal way to communicate.
- Therefore, existing publication templates should include guidelines covering system diagrams.

The lack of guidelines or guidance for diagrams is commonplace. For example, in the "Formatting instructions for ICLR 2021 conference submissions" used in the creation of this paper, there are two pages covering mathematical notation conventions, but no guidance about diagrams beyond a general comment for figures: "You may use color figures. However, it is best for the figure captions and the paper body to make sense if the paper is printed either in black/white or in color." (Rush et al., 2021).

Marshall et al. (2020) use an interview study to derive guidelines for neural network system architecture diagrams. Figure 1 shows visually the impact of these guidelines, which are being evaluated empirically with users (Marshall et al., 2021b), and through a corpus analysis (Marshall et al., 2021a). This proposal does not suggest publisher enforcement of guideline compliance, but would encourage adoption of better practices and highlight to authors the importance of thoughtful diagrams, through the existing peer-review process. Whilst usage of Guidelines, such as those proposed by Marshall et al. (2020), would be an incremental step towards addressing heterogeneity of system diagrams, there is still an opportunity to have further efficiency by implementing more standard notation to more fully address heterogeneity and completeness issues.

### 3.2 POSITION 2: DIAGRAMMATIC SYSTEM SUMMARIES AS STAND-ALONE PUBLICATIONS

This position aims to bring diagrams more centrally as a publication model. Again, we would suggest using the existing LaTeX + PDF workflow for simpler adoption and implementation.

We do not advocate for graphical abstracts, because the majority of literature does not suggest they are effective (see Section 2.3.2). We hypothesise is that in order for complex diagrams to be effective,

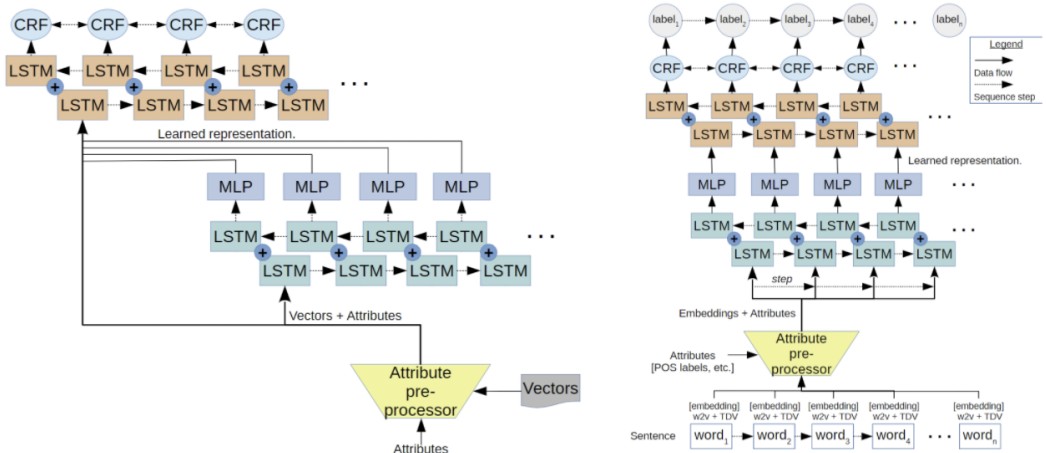

Figure 1: Visual impact of applying guidelines (Group 1, Participant 3). Left: Original, Right: After original author application of guidelines. From Marshall et al. (2021b)

they must be carefully composed, which takes time, skill and experience. We believe that in order for good diagram practices to be widely adopted, the author effort in diagramming ought to be compensated with less work required elsewhere. The guidelines proposed previously would also support the diagram creation.

A diagrammatic system summary would include sufficient information for a peer review within a diagram. Page limit of 2 pages, allowing for an overview diagram and a component explosion or text explanation if required. In this intermediate position, authors can use any visual encoding they wish to express their work, and convince others of their contribution. There is no prescribed formal visual language (as advocated in Position 3). In this way, the authors retain flexibility, whilst having a reduced English communicative burden but increased diagrammatic burden, as the quality and thought required to compose a complete and compelling diagram will be larger than at present. Through analysis of the diagrams as a corpus, there is potential for these flexible visual methods to aid the evolution good practices or standards.

Diagrammatic summaries would allow authors to create papers more quickly. Being shorter and more cognitively proximal to the systems themselves, they would also make it easier for some readers to interpret relational aspects of the system.

This approach would bring substantial attention to diagrams of ML systems, and would be a likely place for emerging standards and conventions around visual encoding of components, as well as the essential information required to communicate ML system research. Further, by maintaining the existing peer review standards, we believe essential information would become apparent very quickly, as reviewers' ability to evaluate publications becomes almost entirely contingent on the interpretability and completeness of the diagram. Longer term, the empirical evaluation of guidelines could provide a scientific basis for decisions about diagram standards.

### 3.2.1 WHY USE THE CURRENT LaTeX + PDF WORKFLOWS?

We do not claim that the current processes are optimal, but we are being pragmatic about adoption by conference organisers. Our proposals can easily be integrated to get the majority of the benefit for users without creating new processes and software to support them, and in doing so creating a pathway to simpler (diagrammatic) publications. This approach is similar to the approach of many conference organisers, where sharing of code and data is encouraged, rather than mandated. In a study of computational linguistics papers, Wieling et al. (2018) found that papers including code are typically more highly cited than those that do not. We would hope a similar situation would occur with standardised, accessible and complete diagrams.

The peer review process associated with LaTeX + PDF adds value by curating claims and providing feedback to authors. Whilst not without issue, this quality check allows for a level of curation that informal dissemination doesn't have. Further, LaTeX + PDF, whether on arXiv or through conferences or journals, is at present a good primary source, and building on that, rather than establishing a competitor (such as a diagram library), is pragmatically motivated.

The key idea with diagrammatic summary publications is to have more easily interpreted, simply comparable, unambiguous diagrams, and the surrounding publication mechanism is less relevant to this position: Our advocacy for diagrams continues to hold for alternative dissemination models such as interactive documents or code repositories.

### 3.3 POSITION 3: A STANDARDISED DIAGRAMMATIC LANGUAGE FOR ML

Our third proposal is more ambitious, in that more coordination is required to make this approach effective. In Position 2, we argue pragmatically for authors creating their own diagrams. In Position 3, we extend to a standardised diagrammatic language for ML.

A standardised diagrammatic language would facilitate synthesis and analysis of successful architectural patterns, with the potential to provide deeper insight into why different architectures perform well. However, for simplicity we advocate starting with consideration of the reader's ease of use.

#### 3.3.1 IMPROVING EASE OF DIAGRAM CREATION AND INTERPRETATION

For readers, a standard for diagrams would afford additional cognitive efficiencies with usage, making it easier and faster to understand how a system works. It would also allow easier comparison between different architectures.

To support adoption, accessibility and ease of use, standard tooling for diagramming that is suitable for existing publication formats would be beneficial, as noted by participants in Marshall et al. (2020). LeNail (2019) has created a diagramming tool, NN-SVG, designed to support diagramming of a limited set of architectures (FCNN, CNN, and one style of Deep Neural Network) for use within the existing publication format. To support diagrammatic summaries adoption and ease of use, a neural network diagramming tool would be highly beneficial. Accessibility, documented by an accessibility statement, would be an important foundation for any tooling.

#### 3.3.2 ADDRESSING HETEROGENEITY

Following information visualisation theory, the manifest heterogeneity is in: What is included (entities, their attributes and relations between them), and how it is visually encoded (visualisation).

Standardised glyphs and specifications for attributes would be an important part of heterogeneity reduction. It would seem useful, in terms of standardising "relations between entities", to adopt a standard visual grammar. Engelhardt & Richards (2020) provide a vocabulary for describing diagrammatic grammar, which could be applied to facilitate consistent structuring of diagrams. Utilisation of this language provides a structured way to discuss and evaluate options during (community creation of) diagram standards.

Conventions will be needed for all aspects of the system, from how to visually represent tensors, to whether common components such as "biLSTM" should be iconic or a text label. This high level architecture is often where the contribution lies, and is a common abstraction level to see in system diagrams, however it is sometimes necessary to visualise changes within components. We are proposing a static representation, so "zooming in" is not possible. Instead, components can be "exploded" to an algorithmic level of detail if required, perhaps following the choices of Olah (2015) which have the versatility and consistency to support both schematic visual and algorithmic description.

#### 3.3.3 IMPROVING COMPLETENESS

By relying on a diagram, completeness will be required for review. This will necessarily include overlaying performance information, perhaps in tables, as seen in DIAL (Marshall & Freitas, 2018). We suggest the diagrammatic summaries be comprised of:

- Title describing the contribution and making the task explicit

- System components, with visual encoding according to a defined specification. Detail of components (e.g. LSTM units) unrolled or exploded in detail only if useful for describing the contribution.

- Input and output

- Datasets, with train/test splits.

- Hyperparameters, including optimal experimental setting ranges if appropriate.

- A set of appropriate performance measures. DIAL embeds this as a table.

- Citations, assembled in a bibliography. Citations may be linked to specific diagram components.

### 3.3.4 DIAL: AN EXAMPLE OF A DIAGRAMMATIC LANGUAGE FOR AI

Adopting a "good" existing style as a starting point would seem sensible, such as those found in influential papers, or in blog posts such as Olah (2015). Another potential starting point is DIAL, a "Diagrammatic AI Language", which attempts to place the contribution of AI system papers into a standard format (Marshall & Freitas, 2018), with low adoption to date. An example DIAL can be seen in Figure 2, including system components together with performance data. We believe DIAL could serve as a starting point for a community effort to define a formal diagrammatic language.

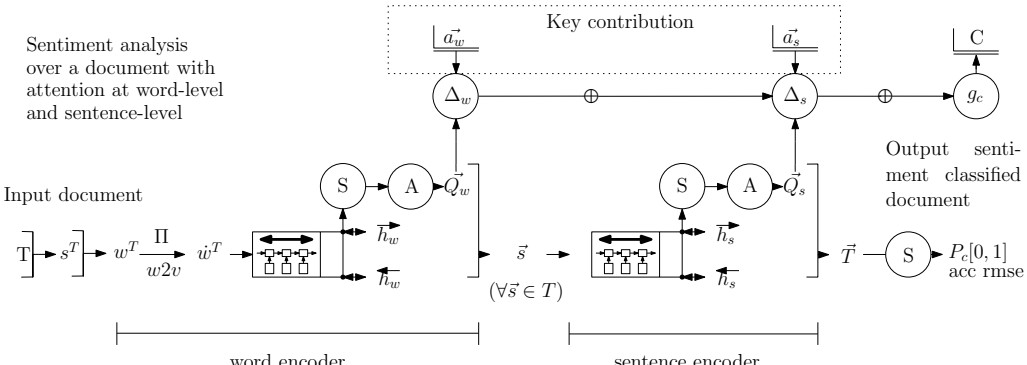

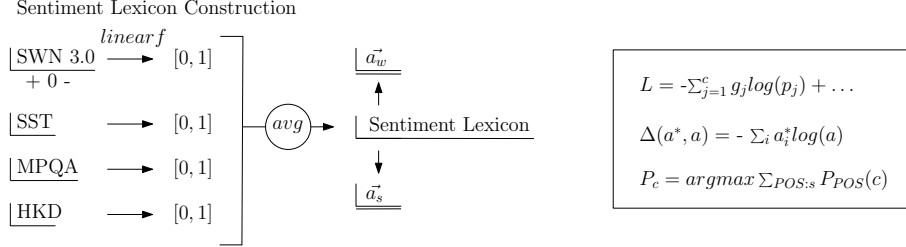

Figure 2: DIAL diagram for a Lexicon-based supervision attention model (Zou et al., 2018), reproduced with permission from the DIAL v0.1 Specification (Marshall & Freitas, 2018)

## 4 REFLECTIONS AND PRACTICAL CONSIDERATIONS

Our proposal utilises existing LaTeX + PDF processes, and allows venue-specified pages for a bibliography of relevant citations, facilitating trivial integration with existing publication processes. In this way, the majority of the logistics such as review cycles, bibliometrics, interoperability, durability, and limitations of static papers would remain unchanged. Particularly with a formal diagrammatic language, there are opportunities to link with innovative publication models, but we omit discussion here to focus on advocating the utility of diagrams in communicating about ML.

Any intervention to improve ML diagrams would benefit from being empirically, rather than solely theoretically, grounded. As noted by Blackwell & Green (2000), different diagram users perform different tasks. To use Blackwell & Green's example, programmers do more restructuring whilst musicians do more search, and any intervention taken here would do well to consider the various educational, research and engineering use cases of ML publications.

Further, in the process of gathering empirical evidence about the properties of diagrammatic representations of ML systems, it is hoped that the quantitative evaluation of the approaches would resonate with ML authors, being a community of data-driven scientists.

## 5 ACCESSIBILITY

As in many areas, there is a risk of cultural bias in diagrams, which can be reduced in order to increase accessibility, participation and engagement. In an intercultural study of students diagrams, Deregowski & Dziurawiec (1986) attributed improper integration of the objects in the figures for the errors, attending elements in isolation. Given the diversity of authors in ML, including the large and increasing scholarly contribution and practical implementation undertaken by authors and institutions of China (Lee, 2018), it is of paramount importance that different cultures be consulted and engaged in the creation of any diagrammatic standards.

This proposal aims to improve accessibility by:

- Adopting an approach of intercultural collaboration from the outset.
- Creating a diverse community to drive diagrammatic standards forward.
- Advocating guidelines which are aligned with existing accessibility checklists, including reducing reliance on colour as noted in Coolidge et al.'s (2018) Checklist for Accessibility.
- Reducing reliance on English language proficiency to communicate about ML systems.

## 6 CONCLUSION

Diagrams are an effective mechanism for communication and dissemination about systems. In this paper we have proposed making diagrams more effective and more central to communication about ML systems. We have highlighted the prevalence of diagrams within the current publication methods (LaTeX + PDF), and the utility of diagrams. Our proposal places diagrams more centrally within the existing formal ML peer review process, either through application of guidelines or the creation of diagrammatic summary publications. Better diagrams have the potential to make communication about ML more effective, complete, and lower effort, reducing load for authors and readers, even within the existing publication processes. Diagrams are an important part of communicating about ML systems, and they should be part of the evolving solution for more efficient and effective communication.

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
