# OpenReview forum: "Diagrammatic summaries for neural architectures"
_ICLR.cc/2021/Workshop/Rethinking_ML_Papers — Rethinking ML Papers - ICLR 2021 workshop Poster_

### Official Review · AnonReviewer2 · 2021-03-30
**Promising problem and discussion but no concrete solution**

**Accessibility:**

Score of 4 (Strong): Submission states accessibility concerns and provides solutions within the proposed framework. However, it does not declare the limitations and exceptions.

**Litreview:**

Score of 3 (Neutral): The submission acknowledges previous work, but does not necessarily explain how the submission differentiates itself (i.e we want to avoid the “deluge of citation” strategy, leaving the reviewer to click through references and figure this part out for themselves).

**Problemstatement:**

Score of 4 (Strong): The submission sets a very strong example of how to address the problem, which should be relevant to the workshop themes.

**Relevance:**

Score of 5 (Exceptional): Like (4) but does so with multiple themes of the workshop.

**Results:**

Score of 3 (Neutral): Submission is well designed and provides a good level of coherency/novelty/interactivity.

**Reviewerconfidence:**

4

My research area is Computer Vision/Graphics which often uses ML networks with complex architecture to perform particular tasks. It is very common (more like a norm) for authors in my area to explain the method pipeline with a diagram for easy understanding, hence I acknowledge the problem discussed in this paper.

**Reviewtext:**

Summary: This paper proposes to improve the explainability/interpretability & visualization of current ML papers by giving more importance to consistent ML system architecture diagrams. The authors propose to have standard guidelines and/or universal language to create consistent diagrams across all ML papers.

Strengths: Uniformity of ML system diagrams across multiple papers is promising next step towards extending the reach of ML papers to the general public, and this topic is very relevant to the workshop theme. The motivation is clearly substantiated with relevant literature. Two of the authors' proposals are reasonable and valid: abiding by standard guidelines for current diagrams and adopting a standard language to create future diagrams.

Weaknesses: The proposals by the authors seem to be a congregation of different ideas (some of them already exist at least partially as cited by the authors) but they don't give a concrete solution to the problem. The success of the proposals are also very conditional; for example, due to difference of opinions and preference in the ML community, agreeing upon a universal standard will be difficult, let alone adopting that standard. I specifically don't think the 2nd proposal of having system diagrams as whole papers is a good idea, because many papers have conceptual novelty rather than just system novelty.

Feedback: I think some preliminary user study of the proposed approaches would have been better to prove their utility and also to compare the proposals among themselves so that we can have more clarity on the solution to the problem.

**Score:**

Accept: The reviewer believes the submission provides a novel and reliable scheme to improve science communication but needs improvement.

---

### Official Review · AnonReviewer1 · 2021-03-30
**Good ideas, proposing too much and does not give enough details on each proposed point**

**Accessibility:**

Score of 3 (Neutral): Submission proposes methods to improve accessibility, but the level of intended accessibility is not well-articulated. Also, the limitations and exceptions are not stated.

**Groundsforrejection:**


Elaboration on point (1) above : typically, in the review process a reviewer may usually ask for improvements of the quality of a certain figure if the figure does not meet the threshold (which is totally subjective).  In other words, how do we set exactly the set of condition under which one can say this diagram is a high quality and that diagram is not ? I think any single metric would rule out many "good" diagrams under some other reasonable metrics. I would more inclined towards a software that helps the authors make very high quality diagrams (and maybe competing software) than the proposed method.


I am afraid to say that I am not in favor of general theme of the paper which advocates for standardization of diagram making (1). The part that seems promising to me is (2) Diagrammatic system summary which advocates for brining diagrams  as the center of a publication model. However the latter does not provide enough details. Finally, part (3) is promising but I cannot advocate for it either because without a functional software, I am just guessing what the authors are trying to convey.

**Litreview:**

Score of 4 (Strong): The submission directly differentiates itself from previous works and formats.

**Problemstatement:**

Score of 2 (Needs Improvement): The submission clearly has potential or credibility, but still fails to state the problem addressed clearly.

**Relevance:**

Score of 4 (Strong): The submission directly addresses a theme of the workshop, and does so in a very professional manner.

**Results:**

Score of 2 (Needs Improvement): Submission shows a poor level of clarity, novelty, coherency, and interactivity.

**Reviewerconfidence:**

4 : I have published multiple papers in visualization and computer graphics venues. Typically in these venues making high quality diagrams is a strict requirement for accepting a paper.

**Reviewtext:**

The article proposes diagrammatic summary for machine learning system architecture papers. The authors suggest three positions:

- Diagram guidelines
- Diagrammatic system summary
- Diagram standards

In general I think the paper has many good ideas (2) and (3) above are in particular promising directions in my opinion. I am afraid to say that the paper really suggest too much without giving each point what it deserves.

(1) For Diagram guidelines the authors mentioned that there is a lack of guidelines for figure in the publishing process. I think I agree with with this point but I think the lack of guidelines is there for good reasons. While there should be a general guideline that the figure must be "high quality" but I think that should be left entirely up to the authors to decide. The think lack of guidelines gives authors more creative ground to make these figures.

(2) Diagrammatic system summary: This idea proposes brining diagrams  as the center of a publication model. I think I am in favor of this part more than (1) and (2). I  wish the author gave examples though about their idea and more details.

(3) Diagram standards: While I appose the idea of making the tool standard, I am in favor of making a tool that helps making the process of diagrams faster and more convenient (provided we do not restrict authors to use that specific tool). Again the danger are the same reasons that made me not in favor of (1) apply here. While an example is provided here but it is too ambitious to propose something like this without a software (see for instance : Penrose: from mathematical notation to beautiful diagrams).




For the above reasons I am sorry that I do not recommend this paper for publication.


Missing from the lit review :

- Penrose: from mathematical notation to beautiful diagrams







**Score:**

Strong reject: The reviewer observed significant issues in the submission.

---

### Meta-Review · Program_Chairs · 2021-04-01

**Recommendation:** Accept
**Confidence:** 4

**Metareview:**

After reading the paper and the two relatively opposing reviews, I think this paper should only be considered as a set of guidelines and not mandates for standardization. However, this paper references two studies either in preparation or review making it hard to see what those studies actually found rather than just the final outputs such as Fig 1.

I agree with reviews in that the authors should have freedom and the reviewers can make suggestions, but standardization might be the way to go. I also agree that the authors should open source a functional software or at least an easy to follow a pipeline with the finer details for researchers to benefit from.

I also agree that the authors should have a user study in this case of subjectivity both for evaluation and creation based on the guidelines.

Overall, the workshop will benefit from the discussions with the authors during the poster session but the authors should further improve upon using the feedback and make it a useful tool/guide.

---

### Decision · Program_Chairs · 2021-04-01

Accept (Poster)